# Laplacian Matrix Learning for Point Cloud Attribute Compression with Ternary Search-Based Adaptive Block Partition

Changhao Peng
pch730@qq.com
Peking University
Shenzhen Graduate School
Shenzhen, China

Wei Gao*
gaowei262@pku.edu.cn
Peking University
Shenzhen Graduate School and Peng Cheng Laboratory
Shenzhen, China

## ABSTRACT

Graph Fourier Transform (GFT) has demonstrated significant effectiveness in point cloud attribute compression task. However, existing graph modeling methods are based on the geometric relationships of the points, which leads to reduced efficiency of graph transforms in cases where the correlation between attributes and geometry is weak. In this paper, we propose a novel graph modeling method based on attribute prediction values. Specifically, we utilize Gaussian priors to model prediction values, then use maximum a posteriori estimation to learn the Laplacian matrix that best fits the prediction values in order to conduct separate graph transforms on prediction values and ground truth values to derive residuals, and subsequently perform quantization and entropy coding on these residuals. Additionally, since the partitioning of point clouds directly affects the coding performance, We design an adaptive block partitioning method based on ternary search, which selects reference points using distance threshold $r$ and performs block partitioning and non-reference point attribute prediction based on these reference points. By conducting ternary search on distance threshold $r$, we rapidly identify the optimal block partitioning strategy. Moreover, we introduce an efficient residual encoding method based on Morton codes for the attributes of reference points while the prediction attributes of non-reference points are modeled using the proposed graph-based modeling approach. Experimental results demonstrate that our method significantly outperforms two attribute compression methods employed by Moving Picture Experts Group (MPEG) in lossless geometry based attribute compression tasks, with an average of 30.57% BD-rate gain compared to Predictive Lifting Transform (PLT), and an average of 33.54% BD-rate gain compared to Region-Adaptive Hierarchical Transform (RAHT), which exhibits significantly improved rate-distortion performance over the current state-of-the-art method based on GFT.

*Corresponding author. This work was supported by The Major Key Project of PCL (PCL2024A02), Natural Science Foundation of China (62271013, 62031013), Guangdong Province Pearl River Talent Program (2021QN020708), Guangdong Basic and Applied Basic Research Foundation (2024A1515010155), Shenzhen Science and Technology Program (JCYJ20230807120808017), and Sponsored by CAAI-MindSpore Open Fund, developed on OpenI Community (CAAIXSJLJJ-2023-MindSpore07).

## CCS CONCEPTS

• **Computing methodologies → Image compression**.

## KEYWORDS

Point cloud compression, adaptive block partition, graph Fourier transform, Laplacian matrix, ternary search, maximum a posteriori estimation.

**ACM Reference Format:**
Changhao Peng and Wei Gao. 2024. Laplacian Matrix Learning for Point Cloud Attribute Compression with Ternary Search-Based Adaptive Block Partition. In *Proceedings of the 32nd ACM International Conference on Multimedia (MM '24), October 28-November 1, 2024, Melbourne, VIC, Australia.* ACM, New York, NY, USA, 9 pages. https://doi.org/10.1145/3664647.3681615

## 1 INTRODUCTION

In recent years, point clouds, as an important representation of three-dimensional data, have attracted increasing attention. Point clouds have broad application prospects in fields such as cultural heritage preservation [5, 30, 52], virtual reality [47], and autonomous driving [51]. In point cloud data, attribute information is an important component for describing object and scene features, such as color and normals. Therefore, effective compression of point cloud attributes is of great significance due to the massive nature of point cloud data. Currently, many existing studies on point cloud compression mainly focus on compressing geometric information [28, 44, 48, 50], such as using different data structures (octree [54], KDtree [19], prediction tree [27], etc.) to compress geometric information, while research on point cloud attribute compression is relatively limited, especially efficient compression methods that maintain data quality require further research and exploration. Therefore, our focus is on lossless geometric-based point cloud attribute compression tasks.

Current traditional point cloud attribute compression methods are based on two basic frameworks: prediction coding [21, 26] and transform coding [9, 25].

The block partitioning method of point clouds largely determines the effectiveness of prediction coding. Regular block partitioning methods aim to divide point clouds into blocks of roughly the same size using basic data structures (octree, KDtree, etc.). The advantage of this approach is that the same block partitioning method can be quickly obtained at the encoding and decoding ends, but there are significant limitations in encoding performance. Existing irregular block partitioning methods [4, 14, 49, 55] require writing the block number of each point into the bitstream, which incurs significant costs when the blocks are small, resulting in decreased encoding performance. We propose a block partitioning method

based on ternary search: first, the point cloud is partitioned based on a distance threshold $r$, and then the optimal distance threshold $r$ is quickly found based on the convexity of the rate-distortion [35] function using ternary search. This allows for dynamic adjustment of block size and position based on the characteristics of point cloud attributes, making compression more compact and accurate. Moreover, only the distance threshold $r$ needs to be transmitted to the decoding end instead of writing the block number of each point into the bitstream, and the decoding end can also quickly obtain the same block partitioning based on the distance threshold $r$.

Discrete cosine transform (DCT) [3] and graph Fourier teansform (GFT) [33] are the main components of point cloud attribute transform coding. DCT does not consider the correlation between geometry and attributes, resulting in poor performance when point clouds have complex textures. The focus of GFT is to construct the Laplacian matrix, and existing methods are based on geometric information to construct the Laplacian matrix [23]. When the correlation between point cloud attributes and geometry is weak, the encoding performance of GFT will significantly decrease. This paper proposes modeling the prediction values of attributes with Gaussian priors [20] and using posterior learning [32] and denoising [8] to construct the optimal Laplacian matrix. Meanwhile, we propose a two-step strategy [11] for quickly solving the non-convex objective functions when constructing the optimal Laplacian matrix.

In this paper, for prediction coding, we utilize a distance threshold $r$ for irregular block partitioning of point clouds and efficiently determine the optimal threshold $r$ by leveraging a ternary search on the rate-distortion function. For transform coding, we model the attribute prediction values utilizing Gaussian priors and subsequently construct an optimal Laplacian matrix based on posterior learning and denoising. In summary, our contributions can be summarized as follows:

- To address the issues of poor performance in regular block coding and the additional bit overhead introduced by block indices in irregular block coding, we propose an adaptive block partitioning strategy based on ternary search to quickly find the optimal block partitioning method of point clouds. Only the distance threshold $r$ needs to be written into the bitstream to quickly restore the block partitioning method at the decoding end.
- To mitigate the decrease in transform efficiency of GFT when the correlation between geometry and attributes is weak, we develop a method to model the prediction values of attributes utilizing Gaussian priors, and then use posterior learning and denoising to construct the optimal Laplacian matrix. Moreover, we propose a two-step strategy for quickly solving the non-convex objective functions.
- Additionally, for the attributes of a small number of reference points generated by distance threshold $r$, we sort their quantized values according to the Morton code in geometry, and then encode the adjacent residuals, fully exploiting the correlation between geometry and attributes.
- Extensive experiments demonstrate that compared with the current state-of-the-art methods based om GFT, the proposed

method can significantly reduce bit overhead while maintaining reconstructed quality, with with an average BD-rate gain compared to MPEG frameworks more than 30%.

## 2 RELATED WORK

### 2.1 Prediction Coding

Prediction encoding is a method that encodes the residuals between prediction values and ground truth [18]. For point clouds, the partitioning method determines the accuracy of attribute prediction values [39]. Therefore, the partitioning method is a critical factor influencing the performance of prediction encoding, primarily categorized into regular partitioning and irregular partitioning methods.

Regular partition methods such as octree or KDtree were used for block partitioning, gradually dividing the point cloud into cubes or bounding boxes with similar numbers of points. Octree [54] is suitable for irregularly sampled point clouds, while KDtree [36] recursively divides the bounding boxes of point clouds according to certain partition ways, such as setting the coordinate axis with the maximum geometric variance as the partition dimension and evenly dividing the points for partitioning to obtain blocks with roughly the same number of points. Schnabel et al. [34] introduce the concept of octrees for the compression of 3D data, providing an efficient framework for partitioning point clouds into octants recursively. Each octant subdivides space into eight equal parts, accommodating points within its volume. Bentley [6] introduce KDtree as a data structure for multidimensional point sets.

Irregular partition methods designed for attribute compression gradually appeared. For example, Xu et al. [49] use K-means [4] to partition point clouds, and Zhao et al. [41, 55] obtain a hierarchical structure of point clouds based on attribute information. The K-means method is used to avoid generating too many isolated points, while the hierarchical structure divides points with similar colors into different layers [41] to exploit the color continuity. Existing irregular partition methods require the encoding end to write the block partitioning method into the bitstream, resulting in additional bit overhead.

### 2.2 Transform Coding

In the field of compression, DCT is widely studied and applied as an important technique. Ahmed et al. [3] introduce the concept of DCT and apply it to image data compression. Watson et al. [46] propose an adaptive JPEG compression algorithm based on DCT, which achieves better compression by dynamically adjusting the quantization step of DCT coefficients. Additionally, Wang et al. [45] propose a hybrid compression algorithm based on wavelet packets and DCT, combining the advantages of wavelet packets and DCT to achieve higher compression ratios and better reconstructed image quality.

GFT is a highly regarded technique in the fields of image processing and image compression. It extends the concept of Fourier transform to the domain of image data, enabling images to be analyzed and processed in the frequency domain. Research on GFT is not limited to the frequency domain representation of images but also includes its applications in the field of compression. GFT is introduced by Shuman et al. [38], who propose a frequency domain

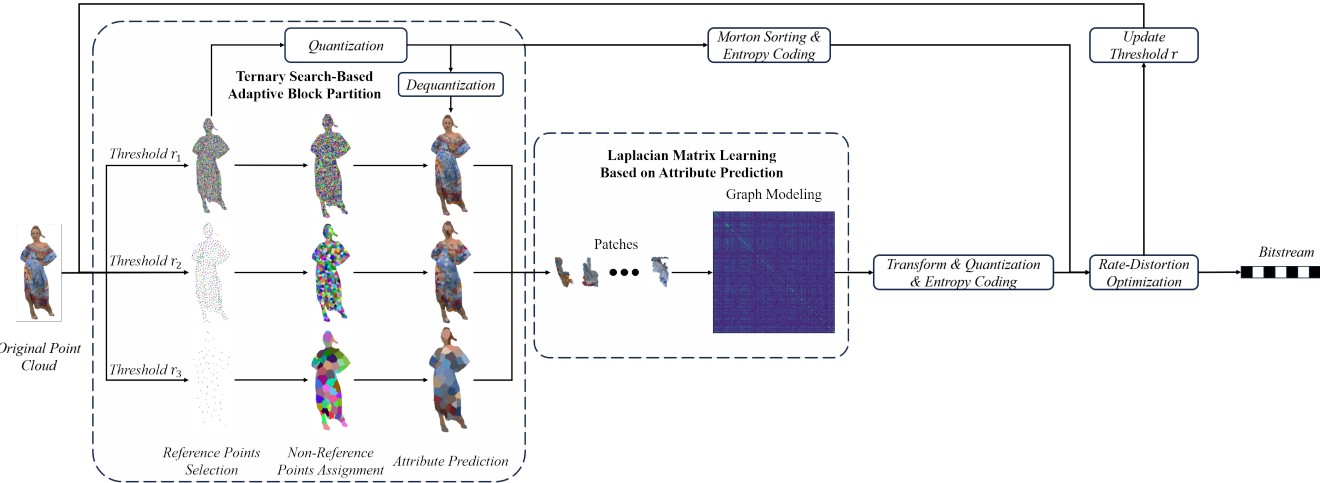

**Figure 1: Overall framework of the proposed method with two main modules. In Ternary Search-Based Adaptive Block Partition, reference points are selected based on a distance threshold $r$, followed by partitioning the point cloud and obtaining attribute predictions for non-reference points. In Laplacian Matrix Learning Based on Attribute Prediction, Gaussian priors are employed to model the attribute predictions of non-reference points, and the optimal Laplacian matrix is constructed through posterior learning and denoising. The attributes of reference points are quantized, sorted according to geometric Morton codes, and then entropy encoded. Finally, the overall rate and distortion are computed, and the distance threshold $r$ is updated accordingly for different patches.**

analysis method based on graph signal processing, treating image data as signals on graphs and utilizing GFT to transform them into the frequency domain for analysis. Fracastoro et al. [13] propose a GFT-based image compression method, leveraging the sparsity of images in the frequency domain, and combining wavelet transform with GFT to achieve efficient image compression. Hu et al. [17] employ GFT for feature extraction and selection in images to reduce data redundancy and achieve compression.

Mekuria et al. [22] use DCT transformation for transform coding of point cloud attributes, which did not utilize the correlation between geometry and attributes. Zhao et al. [56] propose constructing multiple candidate transformation bases and determining the optimal candidate transformation basis using rate-distortion optimization. However, it is difficult to construct suitable candidate transformation bases for different point clouds. Shao et al. [36, 53] use geometric information to construct a graph model and then transformed attributes based on the obtained graph with GFT, effectively utilizing the correlation between point cloud attributes and geometry, but the performance is limited by the point cloud topology. Song et al. [40] propose classifying different blocks for discussion, adopting different mapping methods based on the correlation between attributes and geometry, and using Gaussian Markov random fields [31] for attribute prediction, which to some extent prevents the encoding performance from decreasing due to weak correlation between point cloud attributes and geometry, but still does not fully utilize the correlation of attributes themselves, and excessive classification discussions will bring additional time and bit overhead.

## 3 OUR METHOD

In this section, we give a detailed description of the method we proposed. The overall framework of the proposed method is illustrated

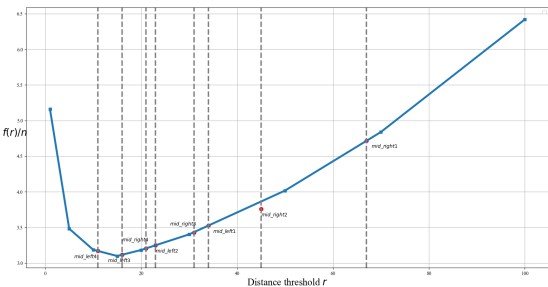

**Figure 2: The ternary search process for point cloud *longdress_vox10_1300.ply,* where the x-axis represents the distance threshold $r$ and the y-axis represents the rate-distortion function $f(r)$ divided by the number of points in *longdress_vox10_1300.ply* (857,966). For the current search range $[l, r]$, the new search range is determined based on the comparison of function values at the left one-third point $mid\_left$ and the right one-third point $mid\_right$. The new search range is two-thirds of the original search range. The search ends when the search range is no larger than the predefined threshold $eps$, and the midpoint of the current search range is outputted as the result.**

in Figure 1. We first introduce the block partitioning method based on the distance threshold $r$, as well as the ternary search algorithm for quickly finding the optimal distance threshold $r$ based on the rate-distortion function. Next, we describe the method of modeling the prediction values of attributes using Gaussian priors and constructing the optimal Laplacian matrix through posterior learning and denoising. Meanwhile we present a two-step strategy for rapidly solving the objective function. Finally, we discuss sorting

the reference points using Morton codes to leverage the correlation between attributes and geometry.

## 3.1 Ternary Search-Based Adaptive Block Partition

**Block Partition with Distance Threshold $r$:** Firstly, we randomly select a point $r_1$ from the original point $P = \{p_1, p_2, ..., p_n\}$ as the first reference point. Assuming that some reference points $R = \{r_1, r_2, ..., r_k\}$ have been selected, then we randomly select a point from the remaining points with a spatial distance greater than the distance threshold $r$ from any reference point as the new reference point $r_{k+1}$. Repeat the above operation until: For any remaining point, there is always a reference point where the spatial distance between the two points is not greater than $r$. At this point, we divide the original point cloud into a reference point cloud $R = \{r_1, r_2, ..., r_l\}$ and a non-reference point cloud $NR = \{nr_1, nr_2, ..., nr_{n-l}\}$.

Next, each non-reference point will be assigned to the nearest reference point, and therefore we divide the original point cloud into $l$ blocks, each containing exactly one reference point. Blocks will serve as the basic unit for subsequent graph modeling and transformations.

Then, we need to predict the attributes of each non-reference point based on the attributes of reference points for subsequent learning of the Laplacian matrix. In light of the fact that both the encoder and decoder must undergo this process, we can only utilize the reconstructed values of reference points' attributes for prediction. The predicted attribute of the non-reference point $nr_i$ is:

$$PA_{nr_i} = \frac{d_2 d_3 RA_{i_1} + d_1 d_3 RA_{i_2} + d_1 d_2 RA_{i_3}}{d_2 d_3 + d_1 d_3 + d_1 d_2}, \tag{1}$$

where $RA_{i_1}$, $RA_{i_2}$ and $RA_{i_3}$ denote the reconstructed values of the three nearest reference points to $nr_i$, while $d_1$, $d_2$ and $d_3$ represent their respective distances to $nr_i$.

**Ternary Search via Rate-Distortion Optimization:** From the above process of block partitioning and prediction, it can be observed that the final encoding outcome is closely related to the selection of the distance threshold $r$: When $r$ is small, more reference points are generated, resulting in lower distortion in the reconstructed point cloud due to the lower quantization level of reference point attributes, albeit at the expense of increased bitstream occupancy. Conversely, when $r$ is relatively large, the distortion in the reconstructed point cloud is higher, but with a smaller bitstream. On the other hand, if we attempt to exhaustively try all values of $r$ (e.g., with a step size of 1), the encoding process would evidently consume a significant amount of time. Therefore, the method for selecting the distance threshold $r$ determines the overall effectiveness and complexity of the encoding process.

One approach is to guide the selection of the distance threshold $r$ based on rate-distortion cost. Our goal is to quickly find the $r$ that minimizes the rate-distortion cost:

$$\min_r f(r) = \mathcal{D}(r) + \gamma_1 \mathcal{R}_R(r) + \gamma_2 \mathcal{R}_{NR}(r),$$
$$s.t.\ r > 0, \tag{2}$$

where $\mathcal{D}(r)$ represents the distortion level of the reconstructed point cloud, denoted by the sum of absolute difference (SAD) or

the sum of square difference (SSD) [37]. $\mathcal{R}_R(r)$ represents the bitstream size of reference point attributes, while $\mathcal{R}_{NR}(r)$ represents the bitstream size of non-reference points. $\gamma_1$ and $\gamma_2$ are tradeoff parameters.

We find that the rate-distortion cost as a function $f(r)$ of the distance threshold $r$ can be approximated by a concave function. For instance, $f(r)$ is illustrated in Figure 2 for *longdress_vox10_1300.ply*.

Considering the property of concave functions, we can utilize ternary search to quickly find the optimal distance threshold $r$ with a complexity of $\mathcal{O}(log_{1.5}(n))$ ($n$ represents the range of $r$). The specific pseudocode is shown in Algorithm 1.

---

**Algorithm 1** Ternary Search via Rate-Distortion Optimization

---

**Input:** Lower bound of the distance threshold $left$; Upper bound of the distance threshold $right$; Precision $eps$; Rate-distortion cost function $f$.
**Output:** The optimal distance threshold $r$.
1: **while** $right - left > eps$ **do**
2:   // Split the search interval into three parts.
3:     $mid1 = left + (right - left)/3$
4:     $mid2 = right - (right - left)/3$
5:     $cost1 = f(mid1)$
6:     $cost2 = f(mid2)$
7:   // Evaluate the function at the two midpoints to determine the
8:   // next search interval.
9:     **if** $cost1 < cost2$ **then**
10:       $right = mid2$
11:     **else**
12:       $left = mid1$
13:     **end if**
14: **end while**
15: // Return the midpoint of the search interval as the result.
16: **return** $(left + right)/2$

---

## 3.2 Laplacian Matrix Learning Based on Attribute Prediction

For a graph $G$, its Laplacian matrix is defined as:

$$L = D - W, \tag{3}$$

where $D$ is the degree matrix of graph $G$, and $W$ is the adjacency matrix. Suppose the Laplacian matrix $L$ has an eigenvalue $\zeta$ and a corresponding eigenvector denoted by $v$. For a signal $x$ on graph $G$, $v^T x$ represents the component of this signal at frequency $\zeta$ under the graph Fourier transform. Different quantization parameters can be applied to coefficients at different frequencies to compress the point cloud, typically with larger quantization parameters for higher-frequency coefficients. Note that $L$ is a real symmetric matrix which can be diagonalized:

$$L = A\Lambda A^T, \tag{4}$$

where the matrix $A$ composed of eigenvectors and $\Lambda$ is a diagonal matrix arranged in ascending order of eigenvalues. $A^T$ can serve as the basis for graph Fourier transform, while $A$ can serve as the basis for the inverse graph Fourier transform.

For $n$ non-reference points within the same block, we obtained the predicted attributes $x \in \mathbb{R}^n$ in the previous step. We aim to learn an appropriate Laplacian matrix based on the predicted attributes $x$. Considering that there is some error (which can be seen as noise) between the predicted attributes and the true attributes, we establish the following model:

$$x = Az + \mu_x + \epsilon, \tag{5}$$

where $z \in \mathbb{R}^n$ is the latent variable corresponding to the true attributes, $\mu_x \in \mathbb{R}^n$ is the mean of $x$, and $\epsilon \in \mathbb{R}^n$ is the noise that follows a multivariate Gaussian distribution [10]:

$$p(\epsilon) \sim \mathcal{N}(0, \sigma_\epsilon^2 I_n). \tag{6}$$

The estimation of latent variable $z$ can be performed using maximum a posteriori (MAP) [15] estimation:

$$z_{MAP}(x) = \arg\max_z p(z|x) = \arg\max_z p(x|z)p(z). \tag{7}$$

For the convenience of subsequent computations, we assume that the latent variable $z$ follows a multivariate Gaussian distribution [12] as below:

$$p(z) \sim \mathcal{N}(0, \Lambda^{-1}), \tag{8}$$

where $\Lambda^{-1}$ is the Moore-Penrose pseudoinverse of $\Lambda$. Therefore, we obtain the following probability distribution:

$$\begin{aligned} p(x|z) &\sim \mathcal{N}(Az + \mu_x, \sigma_\epsilon^2 I_n), \\ p(x) &\sim \mathcal{N}(\mu_x, \Lambda^{-1} + \sigma_\epsilon^2 I_n). \end{aligned} \tag{9}$$

Without loss of generality, we can assume that $\mu_x = 0$. Combining with the probability density function of a $d$-dimensional multivariate Gaussian distribution:

$$f(\mathbf{x}|\boldsymbol{\mu}, \Sigma) = \frac{1}{(2\pi)^{d/2}|\Sigma|^{1/2}} \exp\left(-\frac{1}{2}(\mathbf{x} - \boldsymbol{\mu})^T \Sigma^{-1}(\mathbf{x} - \boldsymbol{\mu})\right). \tag{10}$$

We can learn that:

$$\begin{aligned} z_{MAP}(x) &= \arg\max_z p(z|x) = \arg\max_z p(x|z)p(z) \\ &= \arg\min_z \frac{1}{\sigma_\epsilon^2}||x - Az||_2^2 + z^T \Lambda z \\ &= \arg\min_z ||x - Az||_2^2 + \alpha z^T \Lambda z, \end{aligned} \tag{11}$$

where $\alpha = \sigma_\epsilon^2$ is some constant parameter we can control.

Note that the eigenvalues matrix $\Lambda$, eigenvectors matrix $A$, and latent variable $z$ are all unknowns that we aim to solve for in Equation 11. Therefore, this can be reformulated as the following optimization problem:

$$\min_{\Lambda, A, z} ||x - Az||_2^2 + \alpha z^T \Lambda z. \tag{12}$$

Assuming $y = Az$, combined with Equation 4, we obtain:

$$\min_{L, y} ||x - y||_2^2 + \alpha y^T Ly. \tag{13}$$

Thus, combined with the properties of the Laplacian matrix $L$, we obtained the final objective function and constraints:

$$\begin{aligned} \min_{L, y} &||x - y||_2^2 + \alpha y^T Ly + \beta||L||_F, \\ s.t. \ &tr(L) = n, \ L_{ij} = L_{ji} \leq 0, \forall i \neq j, \ L \cdot \mathbf{1} = \mathbf{0}, \end{aligned} \tag{14}$$

where the Frobenius norm in the objective function controls the connectivity of the graph, the first term of the constraints avoids trivial solutions, and the latter two terms are necessary and sufficient conditions that the Laplacian matrix L must satisfy.

Simultaneously optimizing $L$ and $y$ to solve the objective function 14 is quite challenging. Therefore, we adopt the following two-step strategy.

Firstly, initialize $y$ to be $x$, and then fix $y$ to solve the following problem:

$$\begin{aligned} \min_L &\ \alpha y^T Ly + \beta||L||_F, \\ s.t. \ &tr(L) = n, \ L_{ij} = L_{ji} \leq 0, \forall i \neq j, \ L \cdot \mathbf{1} = \mathbf{0}. \end{aligned} \tag{15}$$

Equation 15 is a standard convex optimization problem that can be solved using existing methods [7]. Then, fix the obtained $L$ and solve the following problem:

$$\min_y ||x - Ly||_2^2 + \alpha y^T Ly. \tag{16}$$

In fact, we can directly obtain an analytical solution to problem 16:

$$y = (I_n + \alpha L)^{-1}x. \tag{17}$$

In this way, we alternately solve for $L$ and $y$ until two consecutive solutions for $y$ satisfy:

$$||y_k - y_{k+1}||_2 < \lambda. \tag{18}$$

The computed Laplacian matrix $L$ obtained through the above calculations can be regarded as the graph structure for the denoised attributes $y$ obtained by denoising the predicted attributes $x$.

Assuming $\hat{x}$ represents the true attributes of these $n$ non-reference points, we then quantify and entropy encode the residuals of the frequency-domain coefficients of the true attributes and predicted attributes:

$$A^T x - A^T \hat{x}, \tag{19}$$

in the graph Fourier transform.

## 3.3 Morton Sorting of Reference Points

For a reference point with spatial coordinates:

$$\begin{aligned} x &= (\overline{x_k x_{k-1}...x_1 x_0})_2, \\ y &= (\overline{y_k y_{k-1}...y_1 y_0})_2, \\ z &= (\overline{z_k z_{k-1}...z_1 z_0})_2, \end{aligned} \tag{20}$$

its corresponding Morton code [29] is:

$$Morton\ code = (\overline{z_k y_k x_k z_{k-1} y_{k-1} x_{k-1}...z_1 y_1 x_1 z_0 y_0 x_0})_2. \tag{21}$$

Utilizing Morton codes derived from three-dimensional geometric coordinates to rank the quantified attributes of reference points:

$$Morton\ order\ attributes = \{n_1, n_2, ..., n_l\}, \tag{22}$$

where $l$ denotes the total number of reference points in the point cloud, and $n_i$ ($i = 1, 2, 3...l$) represents the quantized attributes of the $i$-th reference point after Morton sorting.

Due to the relatively small spatial distances between reference points in small Morton code distance, we can compute the differences between sorted adjacent attributes and subsequently apply entropy coding to these differences:

$$\{n_1, n_2 - n_1, ..., n_l - n_{l-1}\}. \tag{23}$$

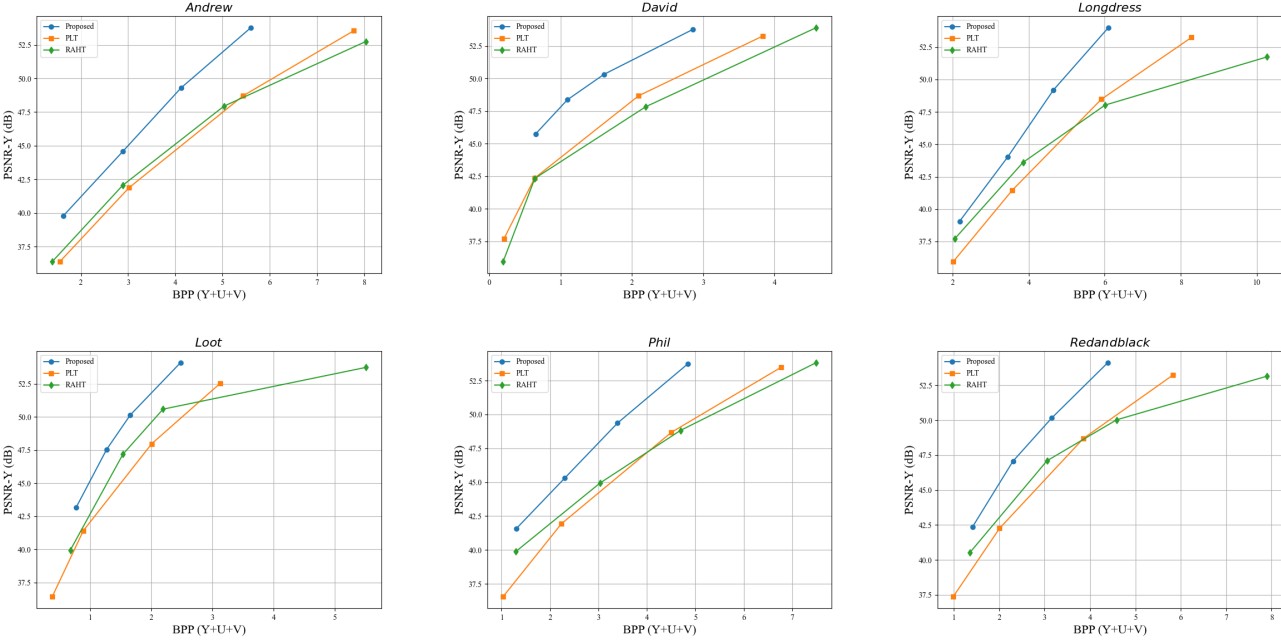

**Figure 3: Demonstration of rate-distortion curves for the proposed method, PLT and RAHT on some of the 9 point cloud sequences.**

## 4 EXPERIMENTAL RESULTS

### 4.1 Experiment Setup

Our experiments are conducted on a PC with Intel Core i7-10510U CPU @ 1.80GHz and 16GB memory. First, we compare our approach with two state-of-the-art frameworks in terms of PSNR and BD-BR performance by the MPEG PCC PSNR [42] evaluation method and the BD-BR performance evaluation tool on traditional video coding [16, 53]. Second, we evaluate the encoding and decoding time of different methods [55]. Finally, we conduct an ablation study to demonstrate the effectiveness of the proposed Laplacian matrix learning method.

**Datasets:** We select nine point cloud sequences from datasets MVUB [24] and 8iVFBv2 [2] for testing. These nine point cloud sequences consist of 200 to 300 frames of point clouds, with each point cloud containing between 500,000 to 2,000,000 points, encompassing point clouds with varying poses and texture complexities. We utilize the average results of each point cloud sequence as the data for comparison.

**Parameter Settings:** We establish the parameter settings based on extensive experimentation and analogous works by others. According to the work of Shao et al. [36], we set the tradeoff parameters in Equation 2 as below:

$$\gamma_1 = \gamma_2 = a \cdot Q^b, \tag{24}$$

where $a = 0.14$, $b = 1.72$ and $Q$ is the quantization parameter. Given that the depth of the point clouds in the test data is consistently 10,

the parameters inputted into Algorithm 1 are:

$$left = 0,$$
$$right = 2^{10} \cdot \sqrt{3} \cdot 0.1 = 102.4\sqrt{3}, \tag{25}$$
$$eps = 0.5.$$

In the computation of parameter $right$, the multiplication by 0.1 is necessitated due to the substantial time consumption in solving the objective function 14 when the number of partitions is insufficiently large, alongside a significant degradation in encoding performance at such junctures.

Finally, we set $\alpha = 10$ and $\beta = 5$ in equation 14 to strike a balance between the fidelity of the graph structure to the signal and the connectivity of the graph structure.

### 4.2 Comparisons with the State-of-the-Arts

To evaluate the effectiveness of the proposed method, we will compare it with two state-of-the-arts point cloud attribute compression methods: (1) Predictive Lifting Transform (PLT) [1], a prediction-based method from Geometry-based Point Cloud Compression (G-PCC version 22) that incorporates both Level_of_Detail (LoD) and lifting transforms; (2) Region-Adaptive Hierarchical Transform (RAHT) [1], a transformation-based method from G-PCC version 22 that utilizes the hierarchical structure of point clouds for inter-layer transforms.

**Rate-Distortion Performance:** We illustrate the relationship in some of the 9 point cloud sequences between the distortion of reconstructed point clouds and the size of the bitstream while utilizing different methods in Figure 3. Herein, we utilize the peak signal-to-noise ratio (PSNR) of the Y component to represent the

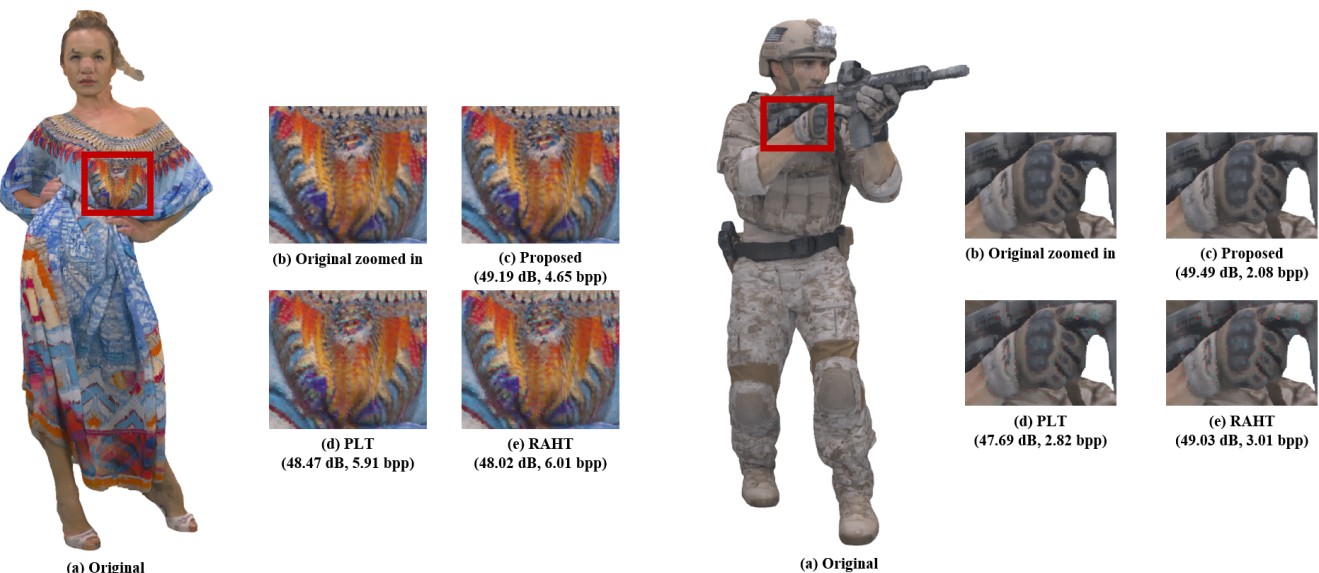

**Figure 4: Visual quality comparison of the reconstructed *longdress* (left) and *soldier* (right) point clouds for proposed method, PLT and RAHT. Each sub-figure is annotated with corresponding bit rate (bpp) and distortion (dB).**

**Table 1: BD-BR (%) comparisons of the proposed method with PLT and RAHT on 9 point cloud sequences.**

| Point Clouds | PLT | | | | | RAHT | | | | | Proposed | |
| | BD-BR (%) | | | Time(s) | | BD-BR(%) | | | Time(s) | | Time(s) | |
| | Y | U | V | Enc. | Dec. | Y | U | V | Enc. | Dec. | Enc. | Dec. |
|---|---|---|---|---|---|---|---|---|---|---|---|---|
| *Andrew* | -27.65 | -19.58 | -15.10 | 35.20 | 32.60 | -27.59 | -22.63 | -21.06 | 36.04 | 34.17 | 174.11 | 29.93 |
| *David* | -41.61 | -49.92 | -13.65 | 42.80 | 40.58 | -51.14 | -47.38 | -22.74 | 46.25 | 43.87 | 254.47 | 44.02 |
| *Longdress* | -23.69 | -21.58 | -17.72 | 22.48 | 21.64 | -20.67 | -16.75 | -18.63 | 25.76 | 24.41 | 82.84 | 16.92 |
| *Loot* | -30.18 | -41.94 | -38.17 | 20.13 | 19.14 | -27.80 | -44.38 | -42.72 | 23.54 | 22.93 | 77.37 | 15.84 |
| *Phil* | -30.60 | -22.20 | -16.99 | 46.77 | 45.03 | -36.88 | -30.84 | -21.50 | 51.32 | 50.19 | 260.06 | 43.18 |
| *Redandblack* | -27.48 | -26.23 | -12.51 | 20.64 | 19.66 | -33.47 | -24.68 | -15.91 | 21.47 | 20.63 | 90.23 | 17.82 |
| *Ricardo* | -36.20 | -42.31 | 23.66 | 26.48 | 24.98 | -37.76 | -27.53 | -5.04 | 29.45 | 29.01 | 117.83 | 26.74 |
| *Sarah* | -25.35 | -28.37 | 7.43 | 40.63 | 38.21 | -38.43 | -32.61 | -15.13 | 46.92 | 41.25 | 177.03 | 38.45 |
| *Soldier* | -32.39 | -45.07 | -38.26 | 28.97 | 27.07 | -28.13 | -41.83 | -37.34 | 30.98 | 29.47 | 130.87 | 25.64 |
| **Average** | **-30.57** | **-33.02** | **-13.48** | **31.57** | **29.88** | **-33.54** | **-32.07** | **-22.23** | **34.64** | **32.88** | **151.65** | **28.73** |

distortion level of the reconstructed point clouds, and bits per point (bpp) to denote the size of the bitstream.

From Figure 3, it can be seen that our method outperforms the other two state-of-the-art reference methods in compression performance across all tested point clouds.

Table 1 presents the BD-BR performance of the proposed method compared to PLT and RAHT on different point clouds, where tests are conducted for the Y, U and V components. Additionally, Table 1 also compares the encoding and decoding time of different methods.Referring to Table 1, compared to PLT, our method achieves an average saving of 30.57%, 33.02%, and 13.48% in bitrates for the Y, U, and V components, respectively, particularly in point clouds *loot* and *soldier* where our method saves over 30% in bitrate for all three components. Compared to RAHT, our method achieves average

savings of 33.54%, 32.07%, and 22.23% in bitrates for the Y, U, and V components, respectively.

Additionally, please note that our results on PLT and RAHT obtained through testing may not be completely consistent with those presented in other papers. This discrepancy may have resulted from inconsistent configuration files or different versions of G-PCC.

**Visual Quality:** In Figure 4, we compare the visual results of different methods on two point clouds *longdress* and *soldier* with complex textures. It can be observed from Figure 4 that, under similar PSNR values for Y component, our method not only saves bitstream compared to PLT and RAHT but also achieves reconstructions more perceptually faithful to complex textures.

**Computational Complexity:** From a complexity perspective, the decoding time of our method is comparable to PLT and RAHT, or even slightly less. However, the encoding time of our method

**Table 2: Ablation study based on block partitioning and transform methods. The results demonstrate the BD-BR (%) comparison between proposed method and comparative methods.**

| Methods | Strategy | | Point Clouds | BD-BR (%) | | |
|---|---|---|---|---|---|---|
| | Partition | Transform | | Y | U | V |
| **Proposed** | Ours | ✓ | - | - | - | - |
| **A** | Ours | × | Andrew | -10.75 | -9.35 | -4.87 |
| | | | David | -15.88 | -17.97 | -5.52 |
| | | | Longdress | -9.80 | -11.12 | -3.47 |
| | | | Loot | -11.46 | -14.59 | -15.99 |
| | | | Phil | -10.82 | -11.51 | -9.81 |
| | | | Redandblack | -10.80 | -12.45 | -5.90 |
| | | | Ricardo | -14.49 | -14.73 | -5.06 |
| | | | Sarah | -12.87 | -13.43 | -5.55 |
| | | | Soldier | -11.54 | -15.92 | -14.93 |
| | | | **Average** | **-12.05** | **-13.45** | **-7.90** |
| **B** | Octree | ✓ | Andrew | -7.57 | -9.29 | -9.13 |
| | | | David | -15.73 | -16.04 | -6.75 |
| | | | Longdress | -10.29 | -9.42 | -8.04 |
| | | | Loot | -13.70 | -14.56 | -19.60 |
| | | | Phil | -13.02 | -12.40 | -9.30 |
| | | | Redandblack | -12.03 | -11.89 | -7.53 |
| | | | Ricardo | -11.12 | -15.33 | -3.80 |
| | | | Sarah | -10.82 | -11.09 | -6.37 |
| | | | Soldier | -14.00 | -19.15 | -13.25 |
| | | | **Average** | **-12.03** | **-13.24** | **-9.31** |
| **C** | Octree | × | Andrew | -15.99 | -12.68 | -12.86 |
| | | | David | -26.16 | -35.17 | -14.71 |
| | | | Longdress | -12.50 | -13.93 | -15.25 |
| | | | Loot | -18.10 | -27.95 | -27.26 |
| | | | Phil | -18.80 | -17.70 | -12.70 |
| | | | Redandblack | -17.63 | -16.38 | -12.90 |
| | | | Ricardo | -24.51 | -23.26 | -8.51 |
| | | | Sarah | -17.18 | -22.72 | -10.29 |
| | | | Soldier | -18.37 | -31.04 | -29.66 |
| | | | **Average** | **-18.80** | **-22.31** | **-16.02** |
| **D** | KDtree | ✓ | Andrew | -11.63 | -7.86 | -4.88 |
| | | | David | -21.75 | -22.65 | -8.92 |
| | | | Longdress | -11.78 | -12.13 | -11.73 |
| | | | Loot | -10.09 | -17.80 | -23.28 |
| | | | Phil | -12.99 | -14.99 | -11.02 |
| | | | Redandblack | -10.66 | -10.27 | -6.01 |
| | | | Ricardo | -17.97 | -17.28 | -0.81 |
| | | | Sarah | -16.11 | -13.18 | -2.03 |
| | | | Soldier | -11.11 | -23.59 | -17.71 |
| | | | **Average** | **-13.79** | **-15.53** | **-9.60** |
| **E** | KDtree | × | Andrew | -21.62 | -11.27 | -11.64 |
| | | | David | -31.41 | -38.72 | -13.01 |
| | | | Longdress | -15.20 | -15.19 | -13.37 |
| | | | Loot | -25.94 | -28.21 | -32.10 |
| | | | Phil | -25.85 | -19.60 | -15.99 |
| | | | Redandblack | -19.24 | -18.73 | -10.98 |
| | | | Ricardo | -23.14 | -30.69 | -8.61 |
| | | | Sarah | -25.77 | -25.21 | -8.37 |
| | | | Soldier | -25.67 | -32.56 | -29.10 |
| | | | **Average** | **-23.76** | **-24.46** | **-15.91** |

is approximately five times that of PLT and RAHT. This is mainly due to the time spent on the ternary search for the optimal distance threshold $r$ and solving Equation 14. Nevertheless, this time expenditure is within acceptable limits, and further optimization of the encoding time can be achieved by utilizing fast algorithm [43] to optimize the solving process of Equation 14.

**Comparison with GFT-based Method:** The work by Song et al. [40] is currently one of the state-of-the-art methods based on GFT for point cloud attribute compression tasks.Since their codes are not available, we compare against the data referenced in the paper for evaluation.

As reported in [40], their method achieves an average bitrate saving of 15.72%, 20.73%, and 23.02% compared to PLT for the Y, U, and V components, respectively, and an average bitrate saving of 13.72%, 22.33%, and 22.94% compared to RAHT. Additionally, this method requires approximately 200 seconds for both encoding and

decoding. Therefore, both in terms of rate-distortion performance and time complexity, our method outperforms theirs.

## 4.3 Ablation Study

In the ablation study, our main objective is to demonstrate the effectiveness of two modules: (1) Ternary search-based adaptive block partition; (2) Laplacian matrix learning based on attribute prediction. Therefore, we set up two variables in the comparative experiments: (1) Partitioning method, where we partition the point clouds based on octree or KDtree instead of ternary search-based adaptive block partition; (2) Removal of the graph transforms, directly quantifying the residuals between the real and predicted attributes (predicted attributes are the weighted averages of the reconstruction values of the three nearest points already encoded in the methods based on octree or KDtree). Therefore, we designed the following five comparative methods:

- Adaptive block partition with no graph transforms.
- Octree-based partition with graph transforms.
- Octree-based partition with no graph transforms.
- KDtree-based partition with graph transforms.
- KDtree-based partition with no graph transforms.

The results of the comparative experiments are shown in Table 2. From Table 2, it can be observed that the performance of the contrast methods exhibits a certain degree of decline compared to the proposed method. The proposed method achieves approximately a 10% to 15% reduction in bitstream compared to the contrast methods that only modify the block partitioning or solely eliminate the transform module. Moreover, compared to the contrast methods that modify both the block partitioning and eliminate the transform module, the proposed method achieves a reduction in bitstream of approximately 20% to 30%. This demonstrates the effectiveness of both Ternary search-based adaptive block partition and Laplacian matrix learning based on attribute prediction.

## 5 CONCLUSION

This paper presents a point cloud attribute compression method based on adaptive block partitioning and GFT. We utilize a distance threshold $r$ for block partitioning of point clouds, and by encoding $r$ into the bitstream, the same block partitioning scheme can be efficiently reconstructed at the decoding end. Simultaneously, we employ a ternary search based on rate-distortion functions to rapidly determine the optimal distance threshold $r$. Furthermore, we propose a novel approach for graph modeling of attribute prediction values, wherein the optimal Laplacian matrix is obtained through posterior learning and denoising, enhancing the transform efficiency of GFT. Experimental results demonstrate that our method achieves an average bitrate reduction of 30.57%, 33.02%, and 13.48% for Y, U, and V components, respectively, compared with PLT. Moreover, compared with RAHT, our method achieves average bitrate reductions of 33.54%, 32.07%, and 22.23% for Y, U, and V components, respectively. The proposed method also exhibits significantly improved rate-distortion performance over the current state-of-the-art method based on GFT. Through ablation experiments, we also validate the effectiveness of the proposed two modules: Ternary search-based adaptive block partition and Laplacian matrix learning based on attribute prediction.

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
