# OpenReview forum: "Laplacian Matrix Learning for Point Cloud Attribute Compression with Ternary Search-Based Adaptive Block Partition"
_acmmm.org/ACMMM/2024/Conference — MM2024 Poster_

### Official Review · Reviewer_BpPK · 2024-05-23

**Rating:** 3
**Confidence:** 3

**Summary:**

This paper presents a point cloud attribute compression method with a ternary search based adaptive block partition and a Laplacian matrix-based optimization.

**Strengths:**

The presented method shows a significant coding gain over the SOTA hand-crafted methods, with an average of 30.57% BD-rate gain compared to Predictive Lifting Transform (PLT), and an average of 33.54% BD-rate gain compared to Region-Adaptive Hierarchical Transform (RAHT).

**Limitations:**

Some clearer explanations of the experimental setup and more sufficient evaluations may be needed.
1) In the Introduction, I cannot clearly see the necessity that the proposed method has to be implemented on the base of lossless geometry coding.
2) In addition, I has some major concerns about the experimental results. First, the version and the configurations of GPCC reference software are not clearly specified in the paper. The rate ranges in the Figure 3 does not match the common rate rages Reviewer has seen in GPCC. It seems the rate points are all relatively higher. Second, in Table 2, when the authors replace or remove the both two proposed key modules, approximately 20% to 30% losses are observed, but still about 10% better than PLT and RAHT if we reference to the Table 1, which confuses Reviewer. The authors may need to explain the source of the gain in these conditions.

**Suitability:**

2

---

### Official Review · Reviewer_kuyR · 2024-05-24

**Rating:** 3
**Confidence:** 3

**Summary:**

This paper proposes a novel graph modeling method based on attribute prediction values. First, the point cloud is divided into reference and non-reference point clouds according to a Distance Threshold 𝑟. Since the value of 𝑟 influences the final encoding result, the optimal 𝑟 is selected through RDO Ternary Search. Maximum a posteriori estimation is then used to learn the Laplacian matrix that best fits the prediction values. This allows for separate graph transforms on the prediction values and ground truth values to derive residuals. These residuals are then subjected to quantization and entropy coding. For the reference point cloud, the attribute residuals are entropy encoded using Morton code sorting.

**Strengths:**

1. This paper proposes partitioning based on Distance Threshold 𝑟 and using Ternary search to select the optimal value of r. From the ablation experiment, compared with Octree and KDTree partitioning, this method will bring good coding gain.
2. The novel approach for graph modeling of attribute prediction values ​​shows good point cloud attribute compression performance compared to RHAT and PLT.

**Limitations:**

1. The subscripts in the formula are confusing. For example, the letter n is used multiple times but represents different meanings. It is not explained what n and l in formula (22) represent respectively.
2. This article mentions in abstract that it is an improvement based on the shortcomings of the existing GFT method, but there is no comparison with other GFT-based point cloud attribute compression methods.
3. Likewise, the GFT-based transformation has not been well compared with the RHAT transformation alone, so we do not know how much gain the new GFT proposed in this method can bring.

**Suitability:**

3

---

### Official Review · Reviewer_y7Uo · 2024-05-24

**Rating:** 4
**Confidence:** 3

**Summary:**

This paper presents a point cloud attribute compression method. It adopts a ternary search based on the rate-distortion function to achieve adaptive block partitioning. To optimize the transformation efficiency of GFT, it uses posterior learning and denoising to obtain the optimal Laplacian matrix.

**Strengths:**

1. This paper proposes an adaptive block partitioning method using a distance threshold r to effectively determine the optimal threshold through ternary search using the rate-distortion function.
2. Experimental results show the superior performance of this method. The authors compare the proposed method with PLT, RAHT and the work of Song et al., showing better results.

**Limitations:**

1. High encoding time. The author says ternary search and objective function 14 are extremely time-consuming. The timing of each intermediate process should be refined.
2. Lack of comparative experiments. The work of Shao et al. is cited in the Experiment Setup but not compared in the evaluation.
3. Lack of analysis on distance threshold r. For example, different r values may have a significant impact on compression performance. No analysis is provided to see such an impact of r.

**Suitability:**

3

---

### Meta-Review · Area_Chair_4hUC · 2024-06-27

**Recommendation:** Accept (Poster)
**Confidence:** 5

**Metareview:**

All reviewers recommend acceptance unanimously.